# Diborides of Multielement Transition Metals: Methods for Calculating Physical and Mechanical Characteristics

Dora Zakarian *, Aik Khachatrian and Sergey Firstov

Frantzevich Institute for Problems of Materials Science, Krzhizhanovsky str., 3, 03142 Kyiv, Ukraine;
khachatryan.h.v@gmail.com (A.K.); sfirstov@ukr.net (S.F.)
* Correspondence: zakarian.d.a@gmail.com

**Abstract:** From the first principles simulation (using the method of "a priori pseudopotential" and the "quasi-harmonic approximation" method- author's developments), the basic characteristics of diborides and diborides of multielement transition metals (DMTMs) with an AlB2 type structure were calculated. For both diborides and DMTMs, the linear coefficients of thermal expansion (LCTE) along the axial axes differ little from each other, i.e., transition metal diborides and hexagonal lattice DMTMs are quasi-isotropic. Quasi-isotropy makes it possible to estimate the LCTE using an analytical formula that depends on the melting temperature. In the absence of experimental data on the melting point of DMTMs, a method for calculating it from first principles is presented. The theoretical hardness values of transition metal diborides and DMTMs with averaged parameters were calculated from the first principles. The hardness of both bulk and nano-sized DMTMs was assessed using a hybrid method. There is agreement between the calculated and available experimental data.

**Keywords:** pseudopotential method; quasi-harmonic approximation; quasi-isotropy; linear coefficient of thermal expansion; theoretical hardness; diborides of multielement transition metals





## 1. Introduction

The interest in studying high-entropy alloys (HEAs) is due to the main advantage of this class of materials—solid solution strengthening. The difference in the atomic radii of the constituent elements leads to distortion of the crystal lattice of the materials and thereby creates a barrier to the movement of dislocations. Therefore, the search for new materials with their inherent high entropy continues.

Among the latest achievements in the field of research of HEAs, one can note the production of high-entropy metal alloys, presented in review articles [1,2]. High-entropy alloys made from refractory metals are potential candidates for high-temperature applications beyond the temperature range of conventional nickel-based superalloys.

Recently, there has been interest in a new group of multicomponent materials—high-entropy transition metal diborides. These materials are not metal alloys but rather metal-like compounds in which metallic bonds between metal atoms coexist with ionic-covalent bonds between metal and non-metal atoms.

If we compare the mechanical properties of metal diborides with those of pure metals, then diborides are characterized by high heat resistance, hardness, elasticity, and the retention of a certain degree of ductility due to the presence of a metallic bond. Experimental data on the hardness of transition metal diborides (4–6 groups) of both bulk and film-type nanomaterials are presented in review articles [3–5]. In the case of bulk materials, the hardness value is in the range of 20–35 GPa, while for thin films it is 40–77 GPa.

If you combine the advantages of HEAs and transition metal diborides, it is possible to obtain alloys with significantly improved physical and chemical properties compared to both those of metal HEAs and those of diborides. Work in this direction is currently being developed, with the main focus devoted to various methods for the synthesis of these materials [6–8].

The purpose of the present work is to calculate from first principles the physical and mechanical characteristics of diborides of multi-element transition metals (crystal lattice parameters, melting point, LCTE, theoretical strength, hardness).

## 2. Theory and Calculation Methods

Unlike the high-entropy compounds of carbides, silicides, and nitrides with a NaCl-type structure, or metal HEAs with a bcc or fcc structure, diborides are characterized by a hexagonal lattice and, therefore, a certain degree of anisotropy in physical properties is inherent.

Ab initio calculations of the LCTE ($\alpha_a$, $\alpha_c$) along the axial axes for transition metals with an hcp lattice differ by a factor of 1.5–1.8, as demonstrated in the case of Zr, $\alpha_c/\alpha_a = 1.74$, and for Ti, $\alpha_c/\alpha_a = 1.6$. Such materials are characterized by high anisotropy along the axial axes.

To assess the hardness or other characteristics of transition metal diborides, it is necessary to determine the degree of their anisotropy. This is especially important in the case of calculating DMTM hardness, as hardness is estimated through a combination of ab initio calculations and the utilization of some experimental data. If the object under study has a cubic structure, then the question of anisotropy does not arise when assessing hardness.

In the calculations, we use the a priori pseudopotential method [9–12]. For DMTMs, the pseudopotential can be represented as

$$\overline{V}(q) = \sum_i C_i V_i(q), \tag{1}$$

where $C_i$ and $V_i(q)$ are the concentration and pseudopotential of the $i$-th element (MeB$_2$). In this work, equiatomic alloys are considered.

In DMTMs, the boron atoms have a "fixed" position in the hexagonal lattice. The nodes of the hexagonal lattice contain metal atoms; the order of arrangement of these atoms can change when moving from one cell to another. If we draw a parallel with metal HEAs, then in DMTMs, it is not the metal atom (Me) but rather the compound (MeB$_2$) that appears as a separate element.

The pseudopotential for transition metal borides MeB$_2$ (Me–V; Cr; Nb; Ta; Hf; Zr; Ti) has the following form:

$$V_{MeB2}(q) = [\Omega_{Me}\, V_{Me}(q) + \Omega_B V_B(q) \sum_{j=1}^{2} \exp(-i\,\vec{q}\,\vec{\delta_j}\,)\,]/\Omega_{MeB2},$$
$$\vec{\delta_1} = \vec{a}\cdot 2/3 + \vec{b}\cdot 1/3 + \vec{c}\cdot 1/2 \;\; ; \qquad \vec{\delta_2} = \vec{a}\cdot 1/3 + \vec{b}\cdot 2/3 + \vec{c}\cdot 1/2 \,, \tag{2}$$

where $\vec{\delta_1}$, $\vec{\delta_2}$ are the radius vectors of boron atoms in the lattice; $\vec{a}$, $\vec{b}$, $\vec{c}$ are the vectors of the hexagonal lattice; $V_{Me}$, $V_B$ and $\Omega_{Me}$, $\Omega_B$ represent the pseudopotential and volume of metal and boron atoms; $\Omega_{MeB2}$–unit cell volume.

The energy of the electron–ion system is calculated in the second order of perturbation theory using the pseudopotential. Based on the minimum energy per group of atoms (MeB$_2$), the cell parameters are then calculated.

Determination of hexagonal lattice parameters: Considering that boron atoms (populated on the (002) plane) are much smaller in size than any transition metal atom, they will have little effect on the distribution of metal atoms on the basal planes. Hence, it is legitimate to assume that, to a first approximation, the basal plane parameter represents the shortest distance between atoms, i.e., $a = 2r_{Me}$, where $r_{Me}$ is the radius of the metal atom located at the nodes of the basal plane of the unit cell.

To determine the parameter $c$, a fixed value of $a$ is taken, and the minimum energy of the electron–ion system is calculated depending on the parameter $c$. The result is the minimum energy of the electron–ion system, denoted as $U(a, c)$, per unit cell with parameters $(a, c)$.

The calculation results are given in Table 1. For transition metal diboride, the calculated values of the lattice parameter are consistent with the experimental data [13]. The correlation (determination) coefficient for lattice parameters is close to 1, which indicates the adequacy of the developed models. From the results obtained, it appears that the ratio $c/a < 1.1$.

**Table 1.** Calculated data on lattice parameters of transition metal diboride.

|  | *a, c* **(nm)** | *a, c* **(nm) exp.** |
|---|---|---|
| $TiB_2$ | 0.3018; 0.3245 | 0.302; 0.322 |
| $ZrB_2$ | 0.3184; 0.3486 | 0.315; 0.353 |
| $HfB_2$ | 0.3166; 0.3423 | 0.317; 0.346 |
| $CrB_2$ | 0.2973; 0.3057 | 0.298; 0.307 |

*2.1. Calculation of Linear Coefficient of Thermal Expansion*

One of the possible ways to calculate the coefficient of thermal expansion is the "quasi-harmonic approximation" method [14], based on determining the dependence of the total energy of the system on the parameters of the crystal lattice at different temperatures.

The total energy of the electron–ion system of a crystalline material can be represented as the sum of the energies of the electron–ion system at temperature T = 0, and the energy of thermal vibrations of ions at temperature T $\neq$ 0. When calculating the energy of the electron–ion system of crystals at zero temperatures, we use the pseudopotential method, and the energy of thermal vibrations can be taken into account using one of the approximate methods—the Debye method or the Einstein method.

When studying the temperature dependence of the physical and mechanical characteristics of crystals, it is more convenient to use the Einstein method, especially when we are talking about systems with a complex structure.

According to the Einstein method, atoms in a crystal lattice vibrate with the same frequency, the value of which is proportional to the force constant of the material.

The force constant is determined using the pseudopotential method through the second derivative of the interatomic interaction energy with respect to the spatial variable. A model has been developed to identify the dependence of the vibration frequency on the volume of the unit cell at different temperatures [14–17].

When calculating the total energy within the harmonic approximation, new force constants and, consequently, frequencies depending on the volume are used, i.e., we obtain the dependence of lattice parameters on temperature.

With the temperature dependence of the lattice parameters, the linear coefficient of thermal expansion (LCTE) can be calculated.

LCTE ($\alpha_a$; $\alpha_c$) is calculated according to the standard method from first principles using the relations:

$$\alpha_a = (a - a_0)/(a_0 \cdot T); \; \alpha_c = (c - c_0)/(c_0 \cdot T). \tag{3}$$

Here, *a* and *c* are the lattice parameters at temperature T (K), while $a_0$ and $c_0$ are the corresponding values at zero temperature. The average LCTE value is estimated using the formula $\alpha = (2\alpha_a + \alpha_c)/3$.

The LCTE values ($\alpha_a$; $\alpha_c$) and their average value are presented in Table 2 at temperature T = 300 K.

**Table 2.** Calculated LCTE of transition metal diborides; experimental values of the average LCTE are presented in parentheses [3,13,18,19].

|  | *a** **(nm)** | $\alpha^* \, 10^6 T^{-1}$ | $T_H$, **K** | $\alpha_a$; $\alpha_c^* \, 10^6 T^{-1}$ | $\alpha^* \, 10^6 T^{-1}$ |
|---|---|---|---|---|---|
| $TiB_2$ | 0.2947 | 4.947 (4.6) | 3500 | 4.80; 5.17 | 4.927 |
| $ZrB_2$ | 0.3128 | 5.365 (5.9) | 3273 | 5.18; 5.60 | 5.320 |
| $HfB_2$ | 0.3097 | 4.860 (5.3) | 3550 | 4.72; 5.10 | 4.841 |
| $CrB_2$ | 0.2860 | 7.630 (6-8) | 2473 | 7.54; 7.75 | 7.609 |

Considering that the parameters of the MeB$_2$ hexagonal lattice and the coefficients of thermal expansion in the directions *a* and *c* differ little from each other, we can accept an approximation in which the hexagonal lattice is considered as cubic, with the parameter $a^* = (0.866c\, a^2)^{1/3}$. Then, the temperature dependence of the LCTE ($\alpha^*$) can be represented by the formula [20]

$$\alpha^*(T) = 0.032 \left( \frac{T}{T_H} \right)^{\frac{1}{4}} / T_H, \tag{4}$$

where $T_H$ is the melting temperature of diborides. The temperature dependence of the LCTE is given in Table 3 for TiB$_2$.

**Table 3.** Calculated LCTE values of TiB$_2$ depending on temperature.

| T, K | $\alpha_a$ $10^6 T^{-1}$ | $\alpha_c$ $10^6 T^{-1}$ | $\alpha^*$ $10^6 T^{-1}$ | $\alpha$ $10^6 T^{-1}$ |
|---|---|---|---|---|
| 100 | 3.61 | 3.91 | 3.75 | 3.78 |
| 300 | 4.75 | 5.10 | 4.94 | 4.91 |
| 500 | 5.08 | 5.53 | 5.62 | 5.44 |
| 700 | 6.00 | 6.54 | 6.14 | 6.23 |
| 900 | 6.31 | 6.87 | 6.51 | 6.59 |
| 1200 | 6.80 | 7.46 | 7.01 | 7.09 |

The average LCTE value is $\alpha \approx \alpha^*$, which means that transition metal diborides with a lattice parameter ratio $c/a < 1.1$ can be considered quasi-isotropic.

In the case of diborides of multielement transition metals, calculations of lattice parameters and LCTE are carried out with pseudopotential (1).

Determination of the parameters of the hexagonal lattice of multielement transition metal diborides from first principles is carried out according to the following scheme:

As a parameter of the basal plane, we take the double value of the average radius of a metal atom, which represents the arithmetic mean of the radii of atoms of all types present in the alloy, taking into account their concentration ratio.

We fix the value of the basal plane parameter (*a*), and calculate the dependence of the total energy of the electro–ionic system on the second parameter (*c*). Based on the minimum energy, we determine the value of the parameter (*c*).

Change the initial value of the parameter (*a*). For each value of the parameter $a_i$, the parameter $c_i$ (corresponding to the minimum energy) is determined.

From the set of energy values, the minimum one with parameters $a_v$, $c_v$ is selected.

According to the calculation results, diborides of multielement metals also exhibit a parameter ratio $c_v/a_v < 1.1$, that is, these alloys can be considered quasi-isotropic. Then, the LCTE ($\alpha_v$) can be estimated using Formula (3), for which it is necessary to calculate its melting temperature in the absence of experimental data.

## 2.2. Melting Point Calculation from First Principles

The theoretical strength is determined through the energy of the electron–ion system U per representative volume; in the case of MeB$_2$, this is the volume of the unit cell. For uniaxial deformations (for example, along the *z* axis), the strength is estimated using the relation

$$\sigma_z = \frac{1}{S\, c} \frac{\partial U}{\partial e_z}, \tag{5}$$

where $e_z$ is the relative strain, $S$ is the area of the atomic plane in a unit cell located perpendicular to the strain axis, and $c$ is the lattice parameter or its projection parallel to the strain axis.

To calculate the melting temperature of materials, we solve the inverse problem. Using materials with a verified melting point as an example, we determine the dependence of the theoretical strength on temperature using the pseudopotential method and the quasi-harmonic approximation [10]. The results of these calculations are presented in [20].

As the calculation results show, for a group of crystalline materials (diborides, transition metal carbides, multi-element alloys), the theoretical strength has a maximum value at temperature T = 0 K and decreases with increasing temperature [20]. At the melting point, for all materials studied, the strength is 84.59–85.4% of the maximum strength.

In [21], it was experimentally proven that boride and metal-ceramic eutectic systems maintain high strength up to a temperature of 0.8 $T_H$ ($T_H$ is the melting temperature of the material), which was confirmed from first principles in both the case of a eutectic system and for individual components [22]. The condition

$$\sigma\left(T_X\right) = 0.85\sigma_0(T = 0) \tag{6}$$

is determined by $T_H = T_X$.

The calculated values of lattice parameters ($a_v$; $c_v$), melting temperature, and LCTE ($\alpha_v$ at T = 300 K) for DMTMs, as calculated through Formula (3), are presented in Table 4.

**Table 4.** Basic characteristics ($a_v$; $c_v$—lattice parameters; $T_H$—melting temperature; $\alpha_v$—LCTE) of multi-element diborides calculated from first principles.

| | $a_v$; $c_v$ (nm) | $T_H$, K | $\alpha_v$ $10^6 T^{-1}$ |
|---|---|---|---|
| $(TiZr)_{0.5}B_2$ | 0.315; 0.341 | 3280 | 5.36 |
| $(TiCr)_{0.5}B_2$ | 0.300; 0.315 | 2910 | 6.23 |
| $(CrZr)_{0.5}B_2$ | 0.309; 0.327 | 2800 | 6.54 |
| $HfZr)_{0.5}B_2$ | 0.318; 0.345 | 3300 | 5.23 |
| $(TiZrHf)_{1/3}B_2$ | 0.313; 0.339 | 3320 | 5.28 |
| $(TiCrHf)_{1/3}B_2$ | 0.310; 0.334 | 2960 | 6.10 |
| $(CrZrHf)_{1/3}B_2$ | 0.310; 0.339 | 2900 | 6.26 |
| $(TiZrCr)_{1/3}B_2$ | 0.306; 0.323 | 2880 | 6.31 |
| $(TiZrHfCr)_{1/4}B_2$ | 0.309; 0.331 | 2950 | 6.13 |

Due to the lack of (published, reliable) experimental data for these quantities, the only way to verify the calculated values of the lattice parameters is by comparison with the corresponding parameters obtained via Vegard's rule. The latter is obtained using the calculated values of the parameters of transition metal diborides (Table 1). The comparison shows that the calculated data are close to the parameter values obtained using the rule of mixtures (Vegard's rule).

Due to the lack of published data on the melting point of DMTMs, the developed method was applied to borides ($MeB_2$ (Me—Ni, Zr, Hf, Cr, ...); $LaB_6$; and eutectic systems such as $LaB_6$—$MeB_2$) with known experimental data on the melting point. There is agreement between the calculated and experimental data [22].

As for DMTMs, there is some information vacuum with respect to both lattice parameters and the melting point. Mainly presented are experimental studies devoted to the synthesis of single-phase, stable alloys. According to our results, DMTMs have the same properties as ordinary diborides, but with averaged physical parameters (which has also been confirmed experimentally). Therefore, the application of the method to determine the melting point of DMTMs is acceptable.

### 2.3. Calculation of Hardness of Transition Metal Diborides

Transition metal diborides exhibit an anomalously high (~97%) elastic recovery of the indentation depth [4,5]. They have a high modulus of elasticity and deform almost elastically, which makes it possible to calculate, from first principles, the theoretical hardness as a property of the surface layer of the material.

When calculating the total energy of the electron–ion system, the influence of the energy of the outer surface of the material is taken into account. To estimate the energy of the outer surface, the method presented in [10] is used. The essence of the method is to redistribute the energy of the electron–ion system (per unit cell) over all faces of the cell.

As a result, the energy density over the outer surface of the unit cell will be:

$$\rho = U/S, \tag{7}$$

where

$$S = 4ac + 2a^2\sqrt{3}, \tag{8}$$

represents the area of the side faces and two basal planes, while $U$ is the total energy of the electron–ion system in the unit cell. According to the method, the energy per one of the base areas (perpendicular to the $c$ axis) will be:

$$\Phi_c = U\left(a^2 0.5\sqrt{3}\right)/(4ac + 2a^2 0.5\sqrt{3}). \tag{9}$$

The coefficient $K_c = 0.5a^2\sqrt{3}/(4ac + 2a^2 0.5\sqrt{3})$ represents the fraction of energy per base area.

If the outer surface of the material coincides with the base plane, then it has half the energy attributable to the base plane with a minus sign, $\Phi_1 = -UK_c/2$.

After this, it is necessary to take into account the influence of the energy of the outer surface on the state of the electron–ion system within the surface layer of the material, which has a thickness, $d$, in the direction of the parameter $c$.

The energy of the electron–ion system in the first cell (adjacent to the outer surface) is represented as: $U_1 = U - UK_c/2$. To calculate the energy of the electron–ion system in the second cell, we use arithmetic averaging of the energies of the first two cells: $U_1 = U - UK_c/2$ and $U$. Then, the energy in the second cell will be: $U_2 = (U_1 + U)/2 = U(1 - K_c/2^2)$, and for the $i$–th cell:

$$U_i = U\left(1 - K_c/2^i\right). \tag{10}$$

We sum $U_i$ for values $i = 1, 2, \ldots j$, where $j = d/c$, and divide by the number $j$. The result is the average energy of the electron–ion system in the surface layer as a function of thickness, $d$:

$$U_c(d) = U(1 - 2K_c \cdot c/d). \tag{11}$$

If in Formula (4) we use the average energy of elementary cells in the surface layer (10), then we ultimately have a functional dependence of the theoretical hardness on the selected material surface thickness in the direction of the axial axes of the hexagonal lattice:

$$H_c(d) = \sigma_c(1 - 2K_c \cdot c/d). \tag{12}$$

Table 4 shows the calculated values of the compressive strength and hardness of transition metal diborides at different thicknesses of the surface layer.

The use of the first-principles method for calculating the theoretical hardness of multielement transition metal diborides is questionable due to the distortion of the crystal lattice. Such a material is far from an ideal structure, and the concept of theoretical hardness disappears. From first principles, such a calculation can be carried out, but with averaged physical parameters and an ideal arrangement of virtual atoms of the same type (having an averaged characteristic).

In this case, the distortion of the crystal lattice, which is the main feature of multi-element alloys, is not directly taken into account. For alloys $(ZrHf)B_2$, $(ZrTi)B_2$, and $(TiZrHf)B_2$ with equiatomic composition, strength and hardness were calculated using the same Formulas (4), (11), and (12) as those for transition metal diborides (Table 5).

**Table 5.** Calculated values of theoretical strength and hardness (in GPa) depending on the thickness of the surface layer (*d*, nm) (the last column presents experimental data on the hardness of thin films with a thickness of 20–40 nm).

| | $\sigma_c$ | $H_c$ | | | | | $H$ [4,5] |
|---|---|---|---|---|---|---|---|
| | | ($d = 1$) | ($d = 3$) | ($d = 5$) | ($d = 10$) | ($d = 20$) | |
| TiB$_2$ | 74.65 | 69.37 | 72.66 | 73.12 | 74.15 | 74.64 | (40–77) |
| ZrB$_2$ | 61.18 | 56.09 | 59.17 | 59.95 | 60.39 | 61.18 | 40 |
| HfB$_2$ | 64.97 | 59.86 | 63.24 | 63.96 | 64.43 | 64.96 | 40 |
| CrB$_2$ | 68.00 | 64.38 | 66.96 | 67.37 | 67.69 | 68.00 | 49 |
| (ZrHf)B$_2$ | 63.12 | 59.85 | 62.03 | 62.47 | 62.79 | 63.12 | |
| (TiZr)B$_2$ | 67.98 | 64.50 | 66.82 | 67.28 | 67.63 | 67.97 | |
| (TiZrHf)B$_2$ | 67.54 | 63.50 | 66.19 | 66.73 | 67.19 | 67.54 | |
| (TiZrHfCr)B$_2$ | 68.05 | 65.00 | 67.03 | 67.44 | 67.75 | 68.05 | |

From the results of a computational experiment, it is observed that diborides of multielement transition metals behave like alloys of the solid solution type.

As is known, the hardness of crystalline materials depends on the interatomic interaction, the size and packing of atoms in the elementary lattice, and the number of valence electrons. Taking these factors into account, the authors of [23,24] presented the hardness of covalent crystals using the formula:

$$H = const \, S_{i,j}/\Omega \,,$$
$$S_{i,j} = \sqrt{e_i e_j}/\left(n_i n_j d_{i,j}\right); \; e_i = z_i/R_i. \tag{13}$$

Here, $n_i$ and $n_j$ represent the coordination numbers of atoms $i$ and $j$, respectively; $d_{i,j}$ is the distance between atoms $i$ and $j$; $z_i$ is the number of valence electrons of atom $i$; $R_i$ is the radius of the ionic core of atom $i$; $\Omega$—volume of the elementary lattice.

For crystals containing several types of atoms, hardness is determined by the formula:

$$H = \frac{\sum_{i=1}^{n} \sqrt{H_{ii}} + \sum_{i \neq j} \sqrt{H_{ij}}}{N_{ij}} = \frac{\sum_{i=1}^{n} \sqrt{H_i H_i} + \sum_{i \neq j} \sqrt{H_i H_j}}{N_{ij}} \,. \tag{14}$$

$H_i$ is the hardness of the material, which is responsible for the interaction of identical atoms, $N_{i,j}$ is the number of possible pair interactions, $n$ is the number of atom types.

With this representation, the hardness of a multi-element alloy appears to be the geometric mean of the hardness of the binary systems involved in the crystal. Due to geometric averaging, the hardness of such a material should be somewhere between the lowest and highest hardness of the system components involved.

Formula (14) does not take into account the size of the dimensional discrepancy. This parameter is important when estimating $H_{i,j}$. Therefore, in Formula (14), we add a factor responsible for distortion. Then, Formula (14) can be represented as:

$$H = 0.5 \left\{ \frac{\sum_{i=1}^{n} \sqrt{H_i H_i}}{n} + \left(1 + \frac{b \, \Delta r}{a_v^*}\right) \frac{\sum_{i \neq j} \sqrt{H_i H_j}}{N_{i \neq j}} \right\}. \tag{15}$$

Here, $\Delta r$ represents the parameter of atomic radius mismatches, and $b$ is to be determined by comparing calculated and experimental data on hardness.

The hardness of multielement transition metal diborides can be calculated using Formulas (13) and (15) based on data from first-principles calculations, and the unknown parameter const is determined by comparing experimental and calculated hardness values. Therefore, in cases where the hardness of transition metal diborides is known experimentally ($H_i$), then these data can be used in Formula (15) to calculate the hardness of the alloy (multi-element transition metal diborides).

To determine the parameter $b$, we use Formula (15) for the $(Zr,Hf)_{0.5}B_2$ alloy. The choice of alloy is not accidental, as the difference between the atomic radii of Zr and Hf is small ($\Delta r = 0.002$ nm), and the lattice parameters of $ZrB_2$ and $HfB_2$ (Table 1) differ little from each other. If relation (15) adequately describes the physical process responsible for the inhibition of dislocations, then we can assume that the hardness of the $(Zr,Hf)_{0.5}B_2$ alloy will be close to the average hardness value of $ZrB_2$ and $HfB_2$, which is confirmed (Table 5). Therefore, we can assume that $b = 1$.

Table 6 shows the calculated values of the hardness of bulk materials, as well as data on the hardness of the alloy estimated using the rule of mixtures ($\overline{H} = \frac{\sum_{i=1}^{n} \sqrt{H_i H_i}}{n}$). For the $(HfZr)_{0.5}B_2$ alloy, $H \approx \overline{H}$ is obtained.

**Table 6.** Calculated values of hardness of diborides of multi-element transition metals (the experimental value is presented in parentheses, obtained using the Vickers method).

| | $(TiZr)_{0.5}B_2$ | $(HfZr)_{0.5}B_2$ | $(TiZrHf)_{1/3}B_2$ | $(TiCrZrHf)_{1/4}B_2$ | $(ZrHfVNbTa)_{1/5}B_2$ |
|---|---|---|---|---|---|
| $\overline{H}$, GPa | 28.50 | 25.50 | 28.33 | 26.45 | 24.52 |
| $H$, GPa | 29.45 | 25.498 | 28.95; (30.22 $\pm$ 2.43) [8] | 31.79 | 26.2; (26) [25] |

To compare theoretical results with experimental findings, the experimental data available in publications are presented.

The hardening of the alloy depends on the ratio of the parameter $\Delta r$ (mismatch parameter) to the average parameter of the alloy crystal lattice $a_v^*$, which is associated with the potentials of different elements, as well as their interaction. Formula (14) takes into account the effect of a distorted crystal lattice on the mechanical properties of the material.

The $H_i$ value ($MeB_2$ hardness) is taken from experimental data for bulk or nano-sized transition metal diborides [4,5].

In the case of nanosized DMTMs, hardness can be calculated using the same Formula (15) if the hardness values of the transition metal diborides included in its composition, with the same shape and size, are known. In Formula (15), the size factor is included into the value of $\overline{H}$.

Superstoichiometric thin films of transition metal diborides with a grain size of ~20 to 40 nm and a hardness of more than 40 GPa in [4] and about 44 GPa in [5] have been experimentally obtained.

Let us estimate the hardness of equiatomic nanoalloys $(Hf,Zr,Ta,V,W)B_2$ using Formula (14), assuming that the average hardness $\overline{H} \approx 40$ GPa (according to experimental data, this is the minimum value for the average hardness of the alloy). The hardness of a film of multielement transition metal diborides with a grain size of 20–40 nm will be $H \approx 42$ GPa. For the same alloy, if $\overline{H} \approx 44$ GPa, then $H \approx 45.76$ GPa, and its experimental value is 45 GPa [7].

## 3. Discussion of the Results

It is generally accepted that diborides of transition metals of groups IV–VII have higher coefficients of thermal expansion along the $c$ axis than along the $a$ axis [26]. This is explained by the presence of a boron network in the base plane, which has strong interatomic bonds.

When calculating the LCTE of transition metal diborides of groups IV–VI, the contribution to the total interaction energy of metal–metal, metal–boron, and boron–boron atoms was assessed. According to the calculation results, the physical parameters of diborides are mainly influenced by the terms responsible for the metal–boron and metal–metal interactions. For transition metal diborides, the effect of the boron–boron interaction on the physical characteristics is negligible due to the small value of the weighting factor, which is the ratio of the volume of a boron atom to the volume of a unit cell. The metal–boron interaction is the same in the $a$ and $c$ directions. The terms responsible for the metal–metal interaction in the same directions differ little from each other (due to the closeness of the values of the parameters $a$ and $c$).

Consequently, we can consider that diborides and alloys of transition metal diborides are quasi-isotropic along the axial axes. This makes it possible to determine the LCTE of high-entropy diborides using Formula (3), which depends on one parameter—the melting temperature. The melting point can be determined from first principles or experimentally.

According to the experimental data, the diborides of multielement transition metals, both in bulk [8,25] and as nanomaterial [6,7], behave like ordinary solid solutions. These theoretical calculations do not contradict this experiment.

The reason for this may be a decrease in the degree of freedom during the formation of distortion of crystal lattices. In diborides, the position of two boron atoms is approximately fixed, which reduces the number of possible random distributions of metal atoms.

The theoretical results obtained for both bulk and nano-sized samples are consistent with this experiment.

## 4. Conclusions

It has been shown that diborides of multi-element transition metals with a hexagonal lattice are quasi-isotropic (the LCTE along the axial axes have similar values). For such materials, the LCTE can be estimated using the same formula as for metals or metal HEAs with a cubic structure, depending on one parameter: the melting temperature.

The melting point of multi-element diborides can be calculated from first principles based on the temperature dependence of the theoretical strength.

DMTM hardness can be estimated for both bulk and nano-sized materials with the same formula, using experimental hardness values of nano or bulk diborides (components).

**Author Contributions:** Conceptualization, D.Z. and A.K.; methodology, D.Z.; software, A.K.; validation, S.F.; formal analysis, D.Z., A.K. and S.F.; writing—original draft preparation, D.Z.; writing—review and editing, A.K. and S.F.; visualization, A.K.; supervision, D.Z.; project administration, D.Z. All authors have read and agreed to the published version of the manuscript.

**Funding:** This research received no external funding.

**Institutional Review Board Statement:** Not applicable.

**Informed Consent Statement:** Not applicable.

**Data Availability Statement:** The original contributions presented in the study are included in the article, further inquiries can be directed to the corresponding author.

**Conflicts of Interest:** The authors declare no conflicts of interest.

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
