# Peer review of "Diborides of Multielement Transition Metals: Methods for Calculating Physical and Mechanical Characteristics"

_2674-0516, doi:10.3390/powders3010004_

Round 1
Reviewer 1 Report
Comments and Suggestions for Authors
All the required comments are listed in the attached file.
please find the attached file

Comments on the Quality of English LanguageAll the required modifications are listed in the attached file.
please find the attached file
Author Response
Dear Reviewer
Thank you very much for the very detailed edits! As for the first remark:
First-principles methods are the general name for quantum mechanical methods that allow one to calculate the physical and mechanical characteristics of materials without the use of experimental information. This expression is a generally accepted terminology in theoretical scientific works and as a rule does not require additional explanation. In this work, pseudopotential calculation of parameters within the framework of first principles was used, as indicated in the abstract. If you think that something additional needs to be specified or clarified, please indicate exactly what.
All other specified edits have been made (file in attachment).
Thank you again for such detailed edits.
With best regards!

Reviewer 2 Report
Comments and Suggestions for Authors
Dear Authority,
The manuscript entitled ‘Diborides of multielement transition metals, methods for cal-2 culating physical and mechanical characteristics’ investigates the basic characteristics of diborides and diborides of multielement transition metals (DMTM) having an AlB2 type structure were calculated. It makes it possible to calculate theoretical melting points and theoretical hardness of MTMD from first principles without conducting experimental studies on the melting point.
I think, the paper includes important information and data which will be useful for literature. However, there are still some points needs to be clarified along with some missing information about this study. It could be considered for publication after minor correction according to following comments/recommendations;
1- It is not stated which experimental hardness values (Vickers, Brinell or Rockwell) are compatible with the theoretical hardness values obtained from first principles. In addition, while comparing the hardness values with the experimental ones in Table 5, the experimental data of only two compositions are given and the experimental data of the other compositions are limited to theoretical.
2- Again, in the study, theoretical melting points were calculated from first principles and their comparison with experimental data was not made in detail. At the same time, the melting points calculated according to first principles need to be compared with the phase diagrams of the compositions. This is the missing aspect of the article.
3- More studies should be cited regarding studies comparing theoretical calculations and experimental studies. In particular, very little reference has been made to the explanation of the theoretical and experimental part of hardness values. Please consider the following articles, especially regarding hardness and phase diagrams.
https://doi.org/10.1016/j.jallcom.2019.151784
https://doi.org/10.31577/km.2022.5.281
The manuscript can be published in Powders after these minor corrections. Best wishes,
Comments on the Quality of English LanguageModerate editing of English language required
Author Response
Dear Reviewer
Thank you very much for the recommendations and questions!
-1. Presented experimental data obtained using the Vickers method. (thank you, I added this information to the text of the article)
Table 5, a comparison of calculation results with experimental data, shows only the experimental data available in publications. Unfortunately, they are not available for all compositions.
-2. Due to the lack of published data on the melting point of DMTM, the developed method was applied to borides (MeB2 (Me - Ni, Zr, Hf, Cr, ......); LaB6 and eutectic systems LaB6 - MeB2) with known experimental data on the melting point. There is agreement between the calculated and experimental data [22].
As for DMTM, there is some information vacuum with respect to both lattice parameters and melting point. Mainly presented are experimental studies devoted to the synthesis of single-phase, stable alloys. According to our results, DMTMs have the same properties as ordinary diborides, but with averaged physical parameters (which is also confirmed experimentally). Therefore, the application of the method to determine the melting point of DMTM is acceptable.
-3. In contrast to the Ni-Alx alloy, single-phase DMTM are considered in this work.
Once again, thank you for your questions, recommendations and comments!
With best regards
Reviewer 3 Report
Comments and Suggestions for Authors
1. Do I understand correctly that the authors used their own developed code? If that's the case, why didn't they use free code? For example, Quantum ESPRESSO or ABINIT? These packages allow you to perform first-principles pseudopotential calculations as well as QHA calculations.
2. Authors need to improve their references to the literature. For example, the potential method. If pseudo-potential of the authors is used, then you need to add links. If a standard potential is used, then you also need to make a link. What type of pseudopotential is used? The QHA method was not developed by the authors; the original work must be referenced so that readers can read it. Link [11] is not available.
3. "energy of thermal vibrations can be taken into account using one of the 104 approximate methods - Debye or Einstein." Why use the Debye model? As I understand it, this approximation is used in the case of low temperatures, and at high temperatures the Einstein approximation is used. Of course, the Debye model also describes high temperatures, but it is a little more complicated. The main difference between these models is in the low temperature range, so it is not clear why two models should be used in the high temperature range.
4. In tables, it is better to use a period as a decimal separator to avoid confusion.
Author Response
Dear Reviewer
Thanks for the recommendations and comments!
1)
Links to the original and English-language publications;
- Pilyankevich A. N., Zakaryan D. A. Model nonlocal pseudopotential;1. Simple metals;2. Diamond and BN;3. Transition metals.Ukr.Phys.Journal. 1985. 30, â„– 12.P. 1861-1865; 1986. 31, â„– 1, 4. P. 93-96; 609-615 (In Russ.).
2. D. Zakarian, Kartuzov V., Khachatrian A. Pseudopotential method for calculating the eutectic temperature and concentration of the components of the B4C – TiB2 , TiB2 - SiC, and B4C – SiC systems / Powder Metallurgy and Metal Ceramics. Springer - 2009. - Vol. 48, â„– 9-10. – P.588-594. 1068-1302/09/0910-0588
3. D. Zakarian, V. Kartuzov, A. Khachatrian, and A. Sayir. Calculation of composition in LaB6–TiB2 and LaB6–ZrB2 eutectics by means of pseudopotential method /Journal of the European Ceramic Society. - 2011. - V.31, â„–7 –P. 1305-1308.
2)
1 Zakarian D. Theoretical Strength of Borides and Quasibinary Boride Eutectics at High Temperatures / D. Zakarian, V. Kartuzov, A. Khachatrian // Powder Metallurgy and Metal Ceramics Springer. - 2015.- Vol. 54, Issue 3. - P.210-214.
- D. Zakarian, V. Kartuzov, A. Khachatrian. Ab-initio calculation of the coefficients of thermal expansion for MeB2 (Me–Ti, Zr) and LaB6 borides and LaB6–MeB2 eutectic composites / Powder Metallurgy and Metal Ceramics. Springer. - 2012.- Vol. 51, â„– 5 - 6. – P. 301-306. (QHA)
3. D. Zakarian, V. Kartuzov, A. Khachatrian. First principles simulation of temperature depandence of the strength in the quasi binary systems LaB6 - MeB2 with tacing into account interfacial interaction / Metal Powder Report, Elsevier. - 2017.- V. 72. - P.- 195-199.
Thanks for the note, all these additional sources have been added to the article. If you are interested, you can also check them out.
3)
When studying the temperature dependence of the physical and mechanical characteristics of crystals, it is more convenient to use the Einstein method, especially when we are talking about systems with a complex structure.
According to Einstein's method, atoms in a crystal lattice vibrate with the same frequency, the value of which is proportional to the force constant of the material.
The force constant is determined using the pseudopotential method through the second derivative of the interatomic interaction energy with respect to the spatial variable. A model has been developed to identify the dependence of the vibration frequency on the volume of the unit cell at different temperatures [5-8].
When calculating the total energy within the harmonic approximation, new force constants and, consequently, frequencies depending on t
4)
Corrected in article
Once again, thank you for your questions, recommendations and comments! With best regards

Round 2
Reviewer 1 Report
Comments and Suggestions for Authors
Accept
Reviewer 3 Report
Comments and Suggestions for Authors
Accept